# The Analysis of Missed Antibiotic De-Escalation Opportunities in Gram-Negative Bloodstream Infections [note 1]

**DOI:** 10.3390/antibiotics14080800

**Published:** 2025-08-06

**Authors:** Mahir Kapmaz, Şiran Keske, Süda Tekin, Özlem Doğan, Pelin İrkören, Nazlı Ataç, Cansel Vatansever, Özgür Albayrak, Zeliha Genç, Bahar Madran, Hanife Ebru Dönmez, Berna Özer, Ekin Deniz Aksu, Defne Başkurt, Metehan Berkkan, Mustafa Güldan, Veli Oğuzalp Bakır, Mehmet Gönen, Füsun Can, Önder Ergönül

**Affiliations:** 1Department of Infectious Diseases and Clinical Microbiology, International School of Medicine, Istanbul Medipol University, 34815 Istanbul, Turkey; 2Department of Infectious Diseases and Clinical Microbiology, Koç University School of Medicine, 34450 Istanbul, Turkey; sirankeske@yahoo.com (Ş.K.); zakbulut@kuh.ku.edu.tr (Z.G.); oergonul@ku.edu.tr (Ö.E.); 3Koç University İşbank Center for Infectious Diseases (KUISCID), 34010 Istanbul, Turkey; natac@ku.edu.tr (N.A.); cvatansever17@ku.edu.tr (C.V.); fucan@ku.edu.tr (F.C.); 4Infectious Diseases and Clinical Microbiology, Ataşehir Acıbadem Hospital, 34642 Istanbul, Turkey; suda.tekin@gmail.com; 5Department of Medical Microbiology, Koç University School of Medicine, 34450 Istanbul, Turkey; ozldogan@ku.edu.tr; 6Department of Public Health, Koç University School of Medicine, 34450 Istanbul, Turkey; bmadran@ku.edu.tr; 7Department of Infectious Diseases and Clinical Microbiology, VKSV American Hospital, 34365 Istanbul, Turkey; 8Koç University School of Medicine, 34450 Istanbul, Turkey; 9Max Planck Institute for Molecular Genetics, 14195 Berlin, Germany; 10Department of Industrial Engineering, Koc University College of Engineering, 34450 Istanbul, Turkey

**Keywords:** de-escalation, antimicrobial de-escalation, neutropenia, ESBL, antimicrobial stewardship, unmet need

## Abstract

**Aim:** Antibiotic de-escalation (ADE) is essential, but appears to be underperformed although being possible, which we refer to as a ‘missed opportunity’. We aimed to analyze the ADE missed opportunities in Gram-negative bloodstream infections (BSIs) in a setting with a high antimicrobial resistance profile. **Methods:** A retrospective, two-centered cohort study was performed from 1 January 2018 to 30 June 2019, including adults with mono- or polymicrobial Gram-negative BSIs. All ADE episodes and 30-day mortality were noted. **Results/Discussion:** Out of 273 BSIs (43 ADE vs. 230 no-ADE episodes), 101 were considered a ‘missed’ opportunity of ADE (36.9%, 101/273). In multivariate analysis, ADE opportunities were missed 4.4 times more (OR = 4.4; 95% CI 1.24–15.9) in the presence of hematological malignancy and 6.2 times more (OR = 6.2; 95% CI 1.76–22.2) in ESBL. Contrary to this, ADE opportunities were missed 0.24 times less (OR = 0.24; 95% CI 0.09–0.61) among patients with *E. coli* BSIs, and 0.17 less (OR = 0.17; 95% CI 0.05–0.67) if ertapenem was used as an empirical agent. The ADE missed opportunity group had a higher mortality rate, which is statistically significant in univariate analysis, but not in multivariate analysis. **Conclusion:** The presence of ESBL and hematological malignancy were the significant barriers to appropriate ADE practice in our study. A good stewardship program must address physician hesitation in ADE practice.

## 1. Introduction

The widely accepted practice of initiating appropriate empirical antibiotics for severe infections has led to the extensive use of broad-spectrum antibiotics. However, antibiotic de-escalation (ADE) appears to be underutilized even when feasible once culture results are available. Despite the Surviving Sepsis Campaign Guidelines advocating for daily ADE assessments in adults with sepsis, its adoption in clinical practice varies [1]. Moreover, one author sarcastically exaggerated ADE as a cherished concept that is more ‘voodoo’ than science due to the lack of well-conducted observational studies and randomized trials supporting its effectiveness [2]. Still, numerous observational and randomized studies concluded that ADE was associated with lower mortality in patients with severe sepsis, septic shock, and ventilator-associated pneumonia [3,4,5,6]. Many infectious diseases (IDs) experts, however, believe that it is not logical to expect a decline in mortality due to ADE. Some authors argue that the link between ADE and lower mortality is most likely due to confounding by indication resulting in significant negative bias [7]. This perspective suggests that clinicians are more likely to implement ADE in patients who are already relatively stable and less critically ill, resulting in lower mortality rates in this group. For patients in more severe conditions, ADE may be avoided even if it is technically possible, potentially reinforcing a correlation between higher mortality and the absence of ADE. Overall, ADE is considered generally safe, may save money, reduces the risk of antimicrobial resistance, and has lower toxicity and side effects [8,9,10,11,12,13].

The prevalence of ADE varies widely in the literature. It is achieved in 35–50% of patients with severe sepsis and ventilator-associated pneumonia, but less than 10% when multidrug-resistant pathogens are present [14]. We believe that ADE is avoided in many daily cases, even when it is possible to apply, which we refer to as an ADE ‘missed opportunity’. In this study, we aimed to assess the variables and the missed opportunities of ADE in Gram-negative bloodstream infections (BSIs) in a setting with a high antimicrobial resistance profile.

## 2. Results

There were a total of 318 patients at bacteremia onset, among whom 19 patients had inappropriate empirical therapy, 30 patients who died within three days of bacteremia, and 10 patients who had ADE applied after seven days who were excluded for more relevant results. Thus, the study continued with patients who underwent ADE within the first 7 days, and those who did not. The number of days from admission to isolation, the duration of hospitalization, the average time between bacterial isolation and the onset of ADE, the time taken to start appropriate therapy following the onset of bacteremia, and the overall length of antibiotic treatment are all given in Table 1. The average time of ADE was determined to be 5.6 days in our study, which is quite close to the seven-day threshold that was established as the cutoff point.

The mean age of the 259 patients with 273 bacteremia attacks was 63.8 (SD:17) years, and 51.6% were males (Table 1). Solid organ tumor (44.3%), diabetes mellitus (25.3%), hematological malignancy (18.7%), neutropenia (16.8%), chronic renal disease (18.7%), chronic obstructive pulmonary disease (4.4%), solid organ transplantation (2.5%), and bone marrow transplantation (autologous and allogeneic) (2.5%) were among the comorbidities for these patients.

The mean duration of antibiotic treatment was 11.9 days (SD: 5.3). During 90 episodes (33%), patients were admitted to intensive care units (ICUs), and during 28 (10%) episodes, mechanical ventilation was applied. A total of 13 (4.7%) episodes were polymicrobial, and 47 (17.2%) episodes had another type of infection other than BSI. *Escherichia coli* (131, 48%), *Klebsiella* spp. (64, 23.4%), *Pseudomonas aeruginosa* (30, 11%), *Acinetobacter* (15, 5.5%), *Enterobacter* spp. (10, 3.6%), *Serratia marcescens* (9, 3.3%), *Stenotrophomonas maltophilia* (4, 1.5%), *Salmonella enteritidis* (4, 1.5%), *Proteus* spp. (4, 1.5%), and others (2, 0.7%) were among the microorganisms.

The secondary bacteremia rate was 237/273 (86.8%). The leading source of bacteremia was urinary (74/273, 27.1%), central catheter-related (40/273, 14.6%), abdominal (45/273, 16.5%), pulmonary (30/273, 11%), and wound (8/273, 2.9%).

Overall 30-day mortality was 23.4% (64/273). The overall ESBLs rates were 28.9% (79/273). Overall, carbapenem resistance was 11.2% (32/273). Overall, colistin resistance was 10.6% (29/273). Among *E. coli* isolates, 25 out of 131 (18.3%) belonged to ST131, a high-risk clone. The ESBL positivity rate in the ST131 clone was 76% (19/25). No high-risk clone was detected among *Klebsiella* spp. and *P. aeruginosa*.

Of 273 bacteremia attacks, 43 ADE episodes were performed (Figure 1). A total of 101 episodes were considered as ADE ‘missed’ (37% (101/273)). No ADE was used in bacteremic episodes caused by carbapenemase-positive isolates and multidrug-resistant *Acinetobacter* sp. (all of which were multidrug-resistant).

ADE practice was significantly associated with 30-day mortality (4/43 in the ADE group vs. 60/230 in the non-ADE group, *p* = 0.017). Missed opportunities were also associated with 30-day mortality (4/43 in the ADE group vs. 26/101 in the missed ADE opportunities group, *p* = 0.026). Table 1 compares bacteremia episodes in patients with ADE, those without ADE (no-ADE), and those who missed an ADE opportunity (missed ADE).

Antibiotic regimens were assigned according to the activity spectrum as imipenem/meropenem > ertapenem/piperacillin + tazobactam > fourth-generation cephalosporins/antipseudomonal third-generation cephalosporins > quinolones/aminoglycoside/third-generation cephalosporins/ureido/carboxy-penicillin > amoxicillin + clavulanic acid > amoxicillin (partly derived from reference [15]). We did not assign a hierarchy between ertapenem and piperacillin–tazobactam nor between quinolones and aminoglycosides. The transition from intravenous to oral administration was included in the definition of ADE. In the case of combination regimens, rank was assigned based on the most potent drug. Switches within a category or no antibiotic switch were referred to as unchanged. Drugs prescribed for less than 24 h were not taken into consideration.

### 2.1. The Comparison of ADE (n = 43) vs. No-ADE (n = 230) Episodes

The factors positively associated with ADE were higher body mass index, community-onset infection, urinary tract infection, empirical use of ertapenem, and the presence of *E.coli* as the causative agent. Factors negatively associated with ADE were the presence of co-morbidities, hematological malignancy, neutropenia, healthcare-associated infection, pulmonary infection, history of hospital stay within the previous three months, need for mechanical ventilation, need for intensive care unit, presence of ESBLs, and carbapenemase (Table 1 and Table 2). In multivariate analysis, the presence of *E.coli*, empirical ertapenem use, ESBLs, and case fatality were significantly associated with ADE (Table 2). The area under the curve (AUC) was calculated to be 80% in the sensitivity analysis of the multivariate model. The fatality was lower in the ADE group. ADE was 3.78 times more likely (OR = 3.78; 95% CI 1.64–8.75) among patients with *E. coli* bacteremia and 3.83 times more likely (OR = 3.83; 95% CI 1.4–10.5) if ertapenem was used as an empirical agent. Lastly, ADE was 0.15 times less likely (OR = 0.15; 95% CI 0.05–0.49) in the presence of ESBLs.

### 2.2. The Comparison of ADE (n = 43) vs. ADE Missed Opportunities (n = 110)

The factors positively associated with ‘missed’ ADE opportunities were lower body mass index, presence of co-morbidities, healthcare-associated infection, history of hospital stay in the last three months, need for ICU, presence of *K. pneumoniae*, and presence of ESBL (Table 1 and Table 3). Missed ADE opportunities were less likely with the empirical use of ertapenem and the presence of *E.coli* as the causative agent. In multivariate analysis, the presence of *E.coli*, hematological malignancy, empirical ertapenem use, presence of ESBL, and case fatality were significantly associated with missed ADE opportunities (Table 3). The AUC was calculated to be 75% in the sensitivity analysis of the multivariate model. Fatality was 2.45 times more likely (OR = 2.45; 95% CI 1.24–15.9) in a missed opportunity than for the ADE group. ADE opportunities were missed 0.24 times less (OR = 0.24; 95% CI 0.09–0.61) among patients with *E. coli* bacteremia and were 0.17 times less likely (OR = 0.17; 95% CI 0.05–0.67) if ertapenem was used as an empirical agent. ADE opportunities were missed 4.4 times more (OR = 4.4; 95% CI 01.24–15.9) if there was a hematological malignancy and 6.2 times more (OR = 6.2; 95% CI 1.76–22.2) in the presence of ESBL.

## 3. Discussion

The frequency of ADE in our cohort was as low as 16%, and ADE opportunity was missed in 37% of the BSI episodes. Hematological malignancy and the presence of ESBLs were two gaps that increase the rate of ADE missed opportunities in our study. If there was a hematological malignancy, ADE opportunities were 4.4 times more likely to be missed (OR = 4.4; 95% CI 1.24–15.9) in our cohort. The implementation of ADE, combined with frequent education and feedback, is reported to effectively reduce inappropriate antibiotic use in febrile neutropenia without compromising patient safety and adverse events [12,16,17]. Tabah et al. recommended ADE to be applied within 24 h of definitive culture results and an antibiogram being obtained in a position statement from a task force of the European Society of Intensive Care Medicine and European Society of Clinical Microbiology and Infectious Diseases (strong recommendation; low quality of evidence) [17]. They suggested that ADE can be applied in neutropenic critically ill patients (moderate recommendation; low quality of evidence). The empirical antibiotics were de-escalated before day 5 in 46 (37%) critically ill hematological patients [18]. The rate of de-escalation was high in that cohort of hematology patients admitted to the ICU and did not appear to have an adverse effect on their outcome. Despite previous studies, we found a high ratio of missed opportunities for ADE in hematology patients. The following explores some reasons clinicians provided for not performing ADE: *‘a) If the patient is complicated, ADE will not be appropriate. b) ADE is not performed at the hematology clinic, and immunosupressed patients such as neutropenic ones do not undergo ADE. c) A winning team should not be changed, so let us not change our initial antibiotic. d) Culture results may not be inclusive enough and what happens if we are too late?.’* We believe it is crucial to thoroughly examine and emphasize both the role and the ratio of ADE in neutropenic patients within antimicrobial stewardship programs.

Despite including patients with bacteremia, we still had a high rate of missed ADE in our study. Most clinicians would agree that treating true bacteremia based on proper culture results is not in doubt. ADE has also been shown to be effective and safe in several studies in bacteremic patients, including monomicrobial [19], community-onset [20,21], and difficult-to-treat Gram-negative bacilli (*Serratia, Pseudomonas*, *Acinetobacter*, *Citrobacter*, or *Enterobacter* sp.) [10]. However, the clinician’s reluctance to de-escalate according to the antibiogram could be due to various factors, including suspicion of co-infection. Therefore, in our study, we accounted for this probability and did not consider ADE ‘missed’ in cases of resistant infection at another site. Nonetheless, additional randomized-controlled trials are needed to confirm the safety and efficacy of the ADE strategy in BSIs.

Furthermore, studies rarely mention and evaluate ADE feasibility cases. In cases judged as patients with de-escalation possibility, ADE was reported to be applied in only 23% of critically ill patients with microbiologically confirmed infections and sepsis or septic shock admitted to ICUs with a high prevalence of antimicrobial resistance. According to Lee et al., only 86 of 189 eligible patients (45%) with septic shock caused by community-onset monomicrobial *Enterobacteriaceae* bacteremia received ADE [20]. There are areas for improvement in implementing the ADE decision-making processes in daily clinical practice.

In our study, the presence of ESBLs decreased the ADE possibility. Antibiotic resistance is one of the leading obstacles limiting ADE practice in healthcare settings. Because of the multi-resistant pathogens in ICUs with high levels of antimicrobial resistance, the feasibility and rates of ADE were limited. However, when ADE was feasible and used, it was reported that it was not associated with increased mortality [22]. On the other hand, we are certainly aware of the intrinsic debate surrounding carbapenem vs. TZP usage in ESBL-positive bacteremic cases [23]. In the MERINO study, the study drug (TZP vs. meropenem against ceftriaxone-resistant *E.coli* or *K. pneumoniae*) was administered at least for the initial four days, followed by discontinuation and a transition to oral treatment as an option. Therefore, in our opinion, studies like MERINO are not a barrier for ADE practice. Our focus was on the outcomes observed within the first seven days of ADE. In our study, in ESBL-producing isolates susceptible to TZP and ertapenem (ETP), cases were considered missed ADEs if no switch from meropenem or imipenem to TZP or ETP was applied despite being feasible. We think that a 7-day course of carbapenem therapy followed by a switch to ETP or TZP, particularly in the presence of clinical and laboratory improvements, should not be viewed with excessive concern.

In the ADE missed opportunity group, we found that fatality was 2.45 times more likely (OR = 2.45; 95% CI 1.24–15.9). According to our understanding, the link between ADE and lower mortality is most likely due to confounding by indication. The clinical stability of the patient favors an easier decision for ADE. However, this confounding factor is rarely addressed in studies. Van Heijl et al. calculated the potential confounder effect of clinical stability on the estimated impact of ADE on mortality in community-acquired pneumonia patients. They also determined that improper clinical stability adjustment results had a significant negative bias [7]. On the other hand, some studies have found a non-significant link between ADE and mortality [24]. ADE was performed more if the ertapenem was started empirically in the presence of *Escherichia coli*. These findings are likely due to a community-onset urinary infection with a low antimicrobial resistance profile, making ADE possible.

Although the timing of ADE is unclear, some studies define the timing as soon as possible when the culture results are available. Others define a time range from 2 to 7 days as the ideal time for ADE [8,21,24,25,26]. In our cohort, we preferred a seven-day range for ADE, which might have caused a higher ADE rate. The average timing of ADE was determined to be 5.6 days in our study, which is quite close to the seven-day threshold that was established as the cutoff point. This highlights the clinicians’ need for additional time to integrate ADE practices effectively into their daily routines.

Teaching programs may hold promise for improved ADE practices. On the other hand, improving ADE, particularly in tertiary care centers, would be difficult. Trupka et al. found that adding an ADE program to a high-intensity daytime staffing model that already practiced a high level of antibiotic stewardship in an academic ICU was not associated with increased ADE or a reduction in overall antibiotic therapy duration [27]. The antibiotic resistance profile, as well as the heterogeneous and complicated nature of the patient population in those settings, may be natural intrinsic barriers to ADE improvement.

The content of this study was partially presented in an oral session [28]. This study has certain limitations, including a heterogeneous patient population concerning BSI sources and antimicrobial stewardship practices across two different hospitals. Additionally, the study involved a relatively modest number of participants. Another limitation is the absence of a widely recognized ADE definition in our study design. Instead, ADE was determined through clinical decision-making, reflecting daily practice. We increased the ADE time window to 7 days in our cohort, which increased the ADE rate. ADE in neutropenic patients and the presence of ESBL and carbapenemase already have intrinsic arguments. There appears to be an unmet need for clinicians to improve their daily practice in these groups. Our study did not include illness severity scores at baseline (e.g., SOFA, APACHE II), which limits the interpretation of mortality differences.

In conclusion, ADE practice needs to be improved, especially in a highly endemic area for resistant Gram-negative infections. The presence of ESBL and hematological malignancy are significant barriers to appropriate ADE practice. Since ADE performance basically depends on a human decision, the reasons from the clinicians for missed ADE opportunities need to be addressed for a more successful stewardship program. Fatality was more likely in ADE missed opportunities, as indicated in the univariate analysis. However, this was not evident in the multivariate analysis. Lower ADE practice in more severe patients could explain why the ADE missed group had a higher mortality rate.

## 4. Materials and Methods

### 4.1. Study Design and Study Population

A retrospective, two-centered cohort study was conducted to determine the feasibility of all ADE opportunities in Gram-negative bloodstream infections (GN-BSIs). We aimed to determine the incidence of all ADE episodes (and also missed opportunities) in various clinical conditions, as well as the microbiological characteristics of patients with bacteremia. The 30-day mortality was also noted.

This study was conducted at the Koç University Hospital (KUH) and the American Hospital (AH) in Istanbul, Turkey. The KUH and AH have the same administrative policy but serve different populations. The KUH is a tertiary private university hospital with 426 beds, including 55 intensive care unit (ICU) beds. Solid organ and bone marrow transplantations are currently being performed at KUH. The AH is a private tertiary hospital with 260 beds, 16 of which are ICU beds. ID consultation is required during the day at KUH but optional at AH. An ID specialist/consultant usually performs a physical bedside review during the consultation.

This retrospective observational study was performed from 1 January 2018 to 30 June 2019. Adults with GN-BSIs from two hospitals were included in the study. All patients who had positive results in at least two blood culture sets were included to avoid contamination. Since the topic of ADE is already open to debate and objections, we aimed to eliminate other factors that could introduce ambiguity. For this reason, despite it leading to a reduction in the number of cases, we preferred to include those with two positive sets. Thus, we aimed especially to rule out patients with positive growth in one bottle/set from blood culture sets obtained from the central venous line. Adult patients with mono- or polymicrobial bacteremia caused by *Enterobacterales* and non-fermentative Gram-negative bacteria were included in the study. Other possible infections besides bacteremia were noted. Non-hospitalized patients and patients who died within the first three days of BSI were excluded. Neutropenia and the presence of extended-spectrum beta-lactamases (ESBLs) and carbapenemase were included. Both community and hospital-acquired BSIs were included.

### 4.2. Microbiological Studies

Isolates were identified using a Vitek 2 system (bioMérieux, Lyon, France) and by MALDI-TOF MS if required in the routine microbiology laboratory. Antibiotic susceptibilities were determined using a Vitek 2 system (bioMérieux, Lyon, France) automated system. Differences in practices between the two hospitals were expected; however, all isolates were later analyzed at the central laboratory in KUH. Meropenem and colistin susceptibilities were re-tested by the broth microdilution method in the laboratory according to EUCAST criteria [29]. Carbapenemase genes of OXA-48, NDM-1, and KPC were determined by multiplex-PCR, and amplicons were sequenced [30]. Genotyping of the *E. coli*, *K. pneumoniae*, and *P. aeruginosa* isolates was carried out by MLST, comparing housekeeping genes according to the protocols published on the Institute of Pasteur website. ST types were determined using Applied Maths Bionumerics V7.6 software.

### 4.3. Definitions

We are aware of the difficulty in defining ADE, but we have defined it according to the literature. The choice of empirical treatment was entirely at the discretion of the primary clinician and or ID (as mentioned above) within their daily routine. ADE is defined as a change to or discontinuation of a drug class that results in a narrower spectrum of coverage until day seven after the onset of a BSI. If ADE is performed after seven days of empirical treatment, it is accepted as ‘not applied.’ In our cohort, we preferred to define ADE within 7 days of initial treatment since definitive antibiogram results for especially resistant Gram-negative bacteria may become available late. Stopping one of the antimicrobial components in the combination is also considered as ADE. If ADE was theoretically feasible but not implemented, this is referred to as a missed ADE opportunity.

In some clinical scenarios, ADE was feasible but not used due to the following reasons: the presence of infection at another site (i.e., intra-abdominal infection), the presence of multiple sources or abscesses, the presence of antibiotic side effects such as allergy or intolerance, or the presence of concern about drug interactions with other drugs and drug penetration issues. In those clinical situations, ADE was NOT considered as ‘missed’ even if it was feasible (Figure 1 and Figure 2). Empirical antibiotic therapy was defined as the drug prescribed before the susceptibility data were available. Appropriate therapy was defined as the treatment regimen that includes at least one active drug, according to EUCAST.

### 4.4. Outcome and Variables

The primary outcomes were mortality and ADE performance. The following data were collected prospectively for all included patients from the time of admission into the ICU: age, sex, body mass index, length of hospital stay, the source of infection, type of bacteria isolated, chronic diseases, comorbidities (including organ and bone marrow transplantation), need for mechanical ventilation, hemodialysis, surgical interventions, and antibiotic therapy. Serum procalcitonin, leukocytes, CRP, and creatinine levels were measured within 24 h of the onset of bacteremia. The percentage of ADE, no ADE, and missed ADE opportunities were recorded.

### 4.5. Statistical Analysis

For descriptive statistics, we examined characteristics among patients who received ADE vs. those who did not. We used an unpaired t-test, Chi-squared test, or Fisher’s exact test. The data were described using proportions and contingency tables for categorical variables and measures of central tendency and dispersion for continuous variables (i.e., age). For multivariate analysis, logistic regression with backward selection was performed, statistical significance was set at <0.05, and STATA 16 (Stata Corp., College Station, TX, USA) was used.

## Figures and Tables

**Figure 1 antibiotics-14-00800-f001:**
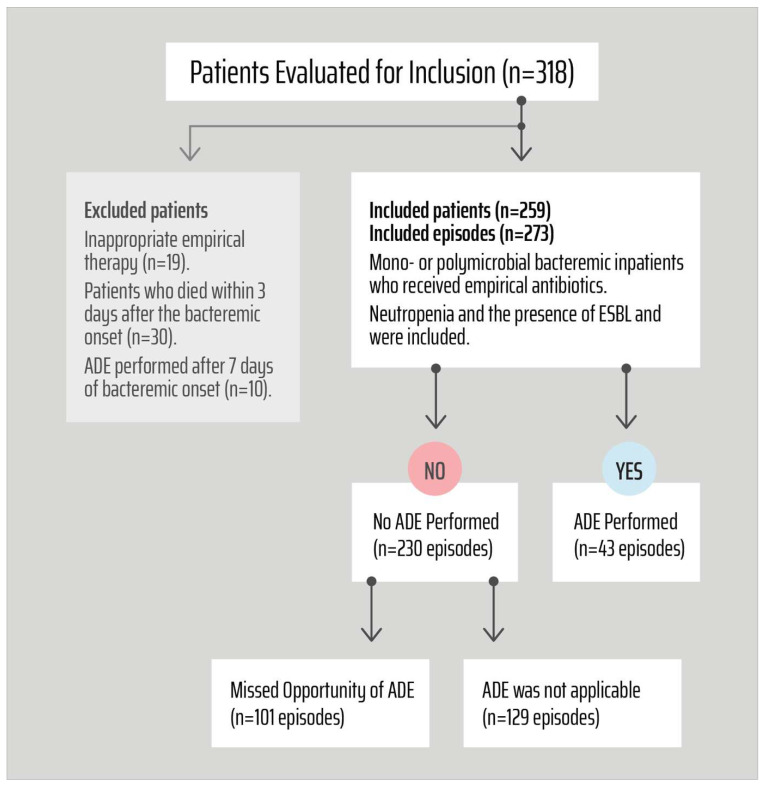
Study population. ADE: Antibiotic de-escalation.

**Figure 2 antibiotics-14-00800-f002:**
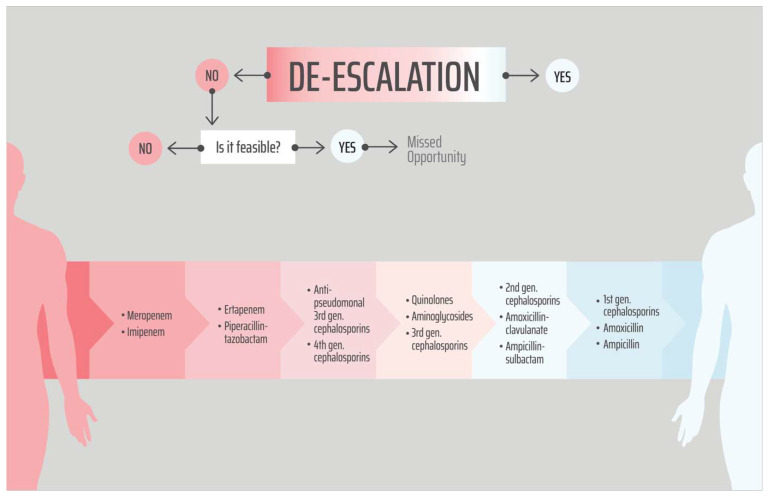
Antibiotic regimens were assigned according to the activity spectrum from highest to lowest. Although no consensual definition of antibiotic de-escalation (ADE) is currently available in the literature, it is established from mainly reference [15]. No hierarchies between ertapenem and piperacillin–tazobactam nor between quinolones and aminoglycosides were assigned.

**Table 1 antibiotics-14-00800-t001:** The comparison of bacteremia episodes among the ADE, no-ADE, and ADE missed opportunity groups.

	De-escalation Performed (ADE) *n* = 43 Episodes (%)	No De-Escalation Performed (No ADE) *n* = 230 Episodes (%)	ADE Missed Opportunity *n* = 101 Episodes (%)	*p* ADE vs. No-ADE	*p* ADE vs. ADE Missed Opportunity
Older age, >65 years	23 (53.4)	133 (57.8)	54 (53.4)	0.598	0.998
Mean age	63.1 (sd: 18)	65.3 (sd: 17)	64.2 (sd: 17.8)	0.456	0.629
Gender, male	17 (59)	124 (54)	55 (54.4)	0.103	0.123
Days from admission to isolation	5.7 (sd: 17)	14.3 (sd: 31)	14.6 (sd: 42.5)	0.089	0.903
Length of stay	31.3 (sd: 117)	32.9 (sd: 39)	32.9 (46.7)	0.865	0.545
Body mass index	27.6 (sd: 6.6)	25.4 (sd: 5.7)	25.4 (sd: 5.7)	0.027	0.024
Days between the isolation of bacteria (day 0) and ADE (mean)	5.1				
Days to initiate appropriate therapy after bacteremia onset	0.39 (sd: 0.1)	0.88 (sd: 1.6)	0.29 (sd: 0.1)	0.059	0.275
Duration of antibiotics therapy	10.6 (sd: 3.5)	13.3 (sd: 6.9)	14.61	0.014	0.999
Comorbidities	34 (79)	216 (94)	98	0.001	<0.001
Diabetes mellitus	9 (21)	60 (26)	28 (27.7)	0.475	0.393
Chronic obstructive pulmonary disease	2 (4.6)	10 (4.3)	6 (5.9)	0.929	0.757
Chronic renal disease	5 (11.6)	46 (20)	18 (17.8)	0.196	0.353
Hematologic malignancy	4 (9.3)	47 (20.4)	25 (24.7)	0.086	0.034
Neutropenia	3 (7)	43 (18.7)	24 (23.7)	0.060	0.018
Solid organ tumor	17 (39.5)	104 (45.2)	41 (40.5)	0.491	0.906
Solid organ transplantation	2 (4.6)	5 (2.1)	2 (2)	0.346	0.372
Bone marrow transplantation (autologous and allogeneic)	0 (0)	7 (3)	4 (4)	0.384	0.353
Healthcare-associated infection	15 (34.9)	144 (62.6)	54 (53.5)	0.001	0.041
Secondary bacteremia	40 (93)	197 (85.6)	84 (83.1)	0.190	0.118
Abdominal	9 (21)	36 (15.6)	14 (13.8)	0.392	0.289
Pulmonary	1 (2.3)	29 (12.6)	10 (9.9)	0.048	0.117
Urinary	19 (44.1)	55 (23.9)	29 (28.7)	0.006	0.071
Catheter-related	4 (9.3)	36 (15.6)	14 (13.8)	0.280	0.449
Wound	0 (0)	8 (3.4)	3 (2.9)	0.214	0.253
History of hospital stay in previous 3 months	19 (44.1)	148 (64.3)	66 (65.3)	0.013	0.018
Mechanical ventilation	0 (0)	28 (12.1)	7 (6.9)	0.016	0.077
Major surgical intervention	14 (32.5)	88 (38.2)	38 (37.6)	0.478	0.562
Transfer to Intensive Care Unit	4 (9.3)	86 (37.3)	35 (34.6)	<0.001	0.002
Polymicrobial bacteremia	2 (4.6)	11 (4.7)	3 (2.9)	0.970	0.614
Simultaneous infections other than bacteremia	3 (6.9)	44 (19.1)	11 (10.9)	0.053	0.468
Laboratory					
Procalcitonin maximum (mg/dl)	15 (sd: 23)	16.6 (sd: 26.4)	19 (sd: 27)	0.710	0.413
CRP maximum (mg/dl)	196 (sd: 96)	197 (sd: 111)	211 (sd: 111)	0.988	0.458
Leucocyte maximum (/μL)	11,759 (sd: 6585)	13,104 (sd: 12,089)	12,716 (sd: 12,556)	0.478	0.683
Creatinine maximum (mg/dl)	1.1 (sd: 0.7)	1.47 (sd: 1.5)	1.5 (1.6)	0.149	0.153
Isolated bacteria					
*Escherichia coli* (*n* = 82)	31 (72)	100 (43.4)	51 (50.5)	0.001	0.017
*Klebsiella pneumoniae* (*n* = 37)	6 (14)	58 (25.2)	31 (30.7)	0.110	0.035
*Acinetobacter* sp (*n* = 15)	0 (0)	15 (6.5)	0 (0)	0.085	NA
*Pseudomonas aeruginosa* (*n* = 11)	1 (2.3)	29 (12.6)	5 (4.9)	0.048	0.471
Empirical use of meropenem	20 (46.5)	127 (55.2)	63 (62.3)	0.293	0.078
Empirical use of ertapenem	12 (28)	17 (7.3)	5 (4.9)	<0.001	<0.001
Empirical use of 3rd gen cephalosporin	1 (2.3)	12 (5.2)	4 (4)	0.414	0.624
Empirical use of piperacillin–tazobactam	10 (23.2)	61 (26.5)	28 (27.7)	0.654	0.578
Presence of ESBL	4 (9.3)	75 (32.6)	28 (27.7)	<0.001	0.015
Presence of carbapenemase	0 (0)	32 (13.9)	0 (0)	0.009	--
Presence of colistin resistance	2 (4.6)	27 (11.7)	0 (0)	0.166	--
MLST type for *E.coli* (ST131)	5 (11.6)	20 (8.7)	13 (12.8)	0.541	0.836
Fatality	4 (9.3)	60 (26)	26 (25.7)	0.017	0.026

**Table 2 antibiotics-14-00800-t002:** Univariate and multivariate analysis for ADE vs. no-ADE (*n* = 259 patients, not episodes).

	**Univariate Analysis**			**Multivariate Analysis**		
	**OR**	**CI**	** *p* **	**OR**	**CI**	** *p* **
Body mass index	1.05	1.01–1.11	0.031	1.05	0.99–1.11	0.104
Healthcare-associated infection	0.32	0.16–0.63	0.001	0.58	0.25–1.32	0.193
*Escherichia coli*	3.36	1.64–6.87	0.001	3.78	1.64–8.75	0.002
Empirical use of ertapenem	4.85	2.11–11.11	<0.001	3.83	1.4–10.5	0.009
ESBL	0.21	0.07–0.62	0.004	0.15	0.05–0.49	0.002
Fatality	0.29	0.1–0.85	0.024	0.36	0.11–1.13	0.08

**Table 3 antibiotics-14-00800-t003:** Univariate and multivariate analysis for ADE vs. ADE missed opportunity (*n* = 137 patients, not episodes).

	**Univariate Analysis**			**Multivariate Analysis**		
	**OR**	**CI**	** *p* **	**OR**	**CI**	** *p* **
*Escherichia coli*	0.4	0.18–0.85	0.018	0.24	0.09–0.61	0.003
Hematological malignancy	3.2	1.05–9.87	0.042	4.4	1.24–15.9	0.022
Empirical use of ertapenem	0.13	0.04–0.41	<0.001	0.17	0.05–0.67	0.011
ESBL	3.7	1.22–11.43	0.021	6.2	1.76–22.2	0.005
Hospital-acquired infection	2.1	1.0–4.5	0.043	0.87	0.34–2.21	0.776
Fatality	3.4	1.1–10.4	0.033	2.45	0.7–8.6	0.161
Prior hospitalization	2.4	1.15–4.93	0.019	1.98	0.85–4.64	0.115

## Data Availability

The data that support the findings of this study are available on request from the corresponding author.

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
