# Peer review of "The Analysis of Missed Antibiotic De-Escalation Opportunities in Gram-Negative Bloodstream Infections [Author-notes fn1-antibiotics-14-00800]"

_antibiotics, 2025, doi:10.3390/antibiotics14080800_

Round 1
Reviewer 1 Report
Comments and Suggestions for Authors
The authors address an important concept of de-escalation of antimicrobial therapy. The manuscript presents data of two centres. Although the novelty of the manuscript is limited, these data might be of interest for the reader.
Methods. Including only patients with two positive blood culture sets is somewhat surprising, as for Gram-negative pathogens contamination is rare. How many cases may have been excluded based on this criterion? Please clarify. How many patients with single positive gram-negative culture were excluded?
Results.
No data on ADE details, e.g. between which drug classes ADE cases occurred, are presented. It would be interesting for the reader -- could the authors add these data? Also timing of ADE would be of interest for the reader -- please add.
As the authors state, definition of ADE is one of the key components of this analysis. The definition is quite clear and appears applicable. However, in the text the authors explain, that ADE after 7 days was defined as missed ADE but on the flow-chart those patients (n=10) were excluded? Please clarify.
In conclusions ther authors state: "The fatality was more likely
in ADE missed opportunities." However, fatality was not different in multivariate analysis between ADE and missed ADE groups? This conclusion seems inappropriate, especially, as the authors themselves state that the mortality difference is most likely confounded by disease severity in many studies. I would suggest rephrasing.
Author Response
Answers to the reviewer 1:
- The authors address an important concept of de-escalation of antimicrobial therapy. The manuscript presents data of two centres. Although the novelty of the manuscript is limited, these data might be of interest for the reader.
- We want to thank you very much for your understanding the main point of this study. In our view, antimicrobial de-escalation (ADE) is an increasingly important issue in the current era of antimicrobial resistance, too.
- Including only patients with two positive blood culture sets is somewhat surprising, as for Gram-negative pathogens contamination is rare. How many cases may have been excluded based on this criterion? Please clarify. How many patients with single positive gram-negative culture were excluded?
- IDSA recomends generally, in adults with a suspicion of BSI, 2 (or sometimes 3) blood cultures (ie, culture sets) should be submitted to the laboratory for the evaluation of each septic episode, with the most important consideration for obtaining more than 2 cultures being the volume of blood sampled (a). They also mention that catheter-drawn blood cultures have a higher risk of contamination (false positives) (a). There are also many epidemics of pseudobacteremia reported for Pseudomonas sp, Burkholderia cepacia, maltophilia (b,c,d).
The topic of ADE is already open to debate and objections. Here, we aimed to eliminate other factors that could introduce ambiguity. For this reason, despite it leading to a reduction in the number of cases, we wanted to include those with two positive sets. Thus, we aimed especially to rule out patients with positive growth in one bottle and/or set from blood culture sets obtained from the CVP line.
During the study design, we did not record the number of single-bottle cultivators. In our future studies, taking this point into consideration and analyzing it would also be beneficial for us.
This point has been incorporated into the text. Thank you for your careful consideration.
- Guide to Utilization of the Microbiology Laboratory for Diagnosis of Infectious Diseases: 2024 Update by IDSA/ASM
- Woo KS, Choi JL, Kim BR, Kim JE, Kim KH, Kim JM, Han JY. Outbreak of pseudomonas oryzihabitans pseudobacteremia related to contaminated equipment in an emergency room of a tertiary hospital in Korea. Infect Chemother. 2014 Mar;46(1):42-4.
- Doit C, Loukil C, Simon A, Ferroni A, Fontan J, Bonacorsi S, Bidet P, Jarlier V, Aujard Y, Beaufils F, Bingen E Outbreak of Burkholderia cepacia Bacteremia in a Pediatric Hospital Due to Contamination of Lipid Emulsion Stoppers. J Clin Microbiol 42:
- del Toro MD, Rodríguez-Bano J, Herrero M, Rivero A, García-Ordoñez MA, Corzo J, Pérez-Cano R; Grupo Andaluz para el Estudio de las Enfermedades Infecciosas. Clinical epidemiology of Stenotrophomonas maltophilia colonization and infection: a multicenter study. Medicine (Baltimore). 2002 May;81(3):228-39.
- No data on ADE details, e.g. between which drug classes ADE cases occurred, are presented. It would be interesting for the reader -- could the authors add these data? Also timing of ADE would be of interest for the reader -- please add.
- In Figure 1, we focused on the hierarchy among the antibiotic classes we outlined. However, we did not conduct a detailed analysis within the groups. In our future research, it would be interesting to investigate which classes undergo switches more frequently.
- Table 1 provides a detailed report of several key metrics: the number of days from admission to isolation, the duration of hospitalization, the average time between bacterial isolation (considered day 0) and the onset of ADE, the time taken to start appropriate therapy following the onset of bacteremia, and the overall length of antibiotic treatment. The average timing of ADE was determined to be 5.6 days in our study, which is quite close to the seven-day threshold that was established as the cutoff point. This highlights the clinicians' need for additional time to integrate ADE practices effectively into their daily routines. This point has been incorporated into the text.
- As the authors state, definition of ADE is one of the key components of this analysis. The definition is quite clear and appears applicable. However, in the text the authors explain, that ADE after 7 days was defined as missed ADE but on the flow-chart those patients (n=10) were excluded? Please clarify.
- We aimed to design our study with a clear and structured approach. We chose to separate 10 patients who underwent ADE after the first 7 days. This decision stemmed from the fact that in the remaining 101 patients, no ADE was performed during their period of 14.6 days of antibiotic treatment. While it would have been possible to group these two categories together, we preferred not to do it. This was entirely our choice, and we highlighted it in the results part. Thank you again for your support.
- In conclusions ther authors state: "The fatality was more likely in ADE missed opportunities." However, fatality was not different in multivariate analysis between ADE and missed ADE groups? This conclusion seems inappropriate, especially, as the authors themselves state that the mortality difference is most likely confounded by disease severity in many studies. I would suggest rephrasing.
- Thank you for your consideration. We have made the changes accordingly:.. The fatality was more likely in ADE missed opportunities, as indicated in the univariate analysis, although this was not evident in the multivariate analysis… We wanted to highlight the fatality-ADE issue in the last paragraph, as it can often be challenging to grasp the underlying bias for the common reader.
Reviewer 2 Report
Comments and Suggestions for Authors
-Thanks a lot for revisiting this importance topic of ADE in a time that rational use of antimicrobials is a global concern and agenda.
-I realized that the last 6 months of the study coinceded with begining of COVID-19 but you did not mention the impact of this pandemic on the wide and unsolicited use of all kinds of antibiotics specially as we 90 percent of intensive care units admissions were covid19 related.
-It could have been an added value if you compare the microbiology lab results between the 2 involved hospitals as one of them using Vitek2 and the othe MALDI-TOFF MS with the known differences in sensitivity.
-The different roles of the internist,the ID physcian and the presence of standing IPC committee need to be discussed with regard to ADE.
-In my view the study could have reached more better results if a PICO approach is adopted but never the less the obtained results were relevant.
-
Author Response
Answers to the reviewer 2:
- -Thanks a lot for revisiting this importance topic of ADE in a time that rational use of antimicrobials is a global concern and agenda.
- We sincerely thank you for giving this topic the importance it deserves.
- I realized that the last 6 months of the study coinceded with begining of COVID-19 but you did not mention the impact of this pandemic on the wide and unsolicited use of all kinds of antibiotics specially as we 90 percent of intensive care units admissions were covid19 related.
This retrospective observational study was conducted between January 1, 2018, and June 30, 2019. Since COVID-19 was officially reported in Turkey in March 2020, the study did not address any aspects related to the pandemic.
- It could have been an added value if you compare the microbiology lab results between the 2 involved hospitals as one of them using Vitek2 and the othe MALDI-TOFF MS with the known differences in sensitivity.
- Differences in practices between the two hospitals were expected; however, all isolates were later analyzed at the central laboratory in KUH. Unfortunately, we did not document the differences between the hospitals not just in terms of microbiological factors but also with respect to ADE practices. It could be interesting to explore the behavioral differences between hospitals, whether private, foundation-run, or public. We included a phrase about this in the section on microbiological analysis. Thank you.
- The different roles of the internist, the ID physcian and the presence of standing IPC committee need to be discussed with regard to ADE.
- This aspect is undoubtedly the most critical component of ADE practices. However, the human behavioral factors involved lie beyond the scope of this current s The complexities of these interactions warrant further investigation in future research efforts. In this study, we focused on examining ADE failure within daily routines. Future research should aim to explore the underlying reasons behind these failures. Thank you for your comment.
- In my view the study could have reached more better results if a PICO approach is adopted but never the less the obtained results were relevant.
- Thank you for your valuable consideration. We could not use PICO this time.
Reviewer 3 Report
Comments and Suggestions for Authors
- The study does not include illness severity scores at baseline (e.g. SOFA, APACHE II), which limits interpretation of mortality differences. Could higher mortality in the non-ADE group reflect more severe illness rather than the absence of ADE?
- In a patient with ESBL-producing E. coli bloodstream infection who was started on meropenem and remained hemodynamically unstable (e.g. on vasopressors) with hypoalbuminemia, clinicians might avoid switching to ertapenem despite in vitro susceptibility, due to concerns about suboptimal pharmacokinetics in critical illness. In such cases, was this considered a missed ADE?
- In ESBL-producing isolates susceptible to piperacillin-tazobactam, were cases that remained on carbapenem classified as missed ADE? Given MERINO data suggesting inferior outcomes with piperacillin-tazobactam compared to meropenem potentially due to OXA-1 co-production—was the lack of de-escalation in this study setting influenced by low OXA-1 prevalence?
- There is no discussion on whether the antibiotics used were appropriate, especially in cases with carbapenemase-producing organisms. This could have influenced mortality, regardless of ADE decisions.
- There is no discussion on whether the antibiotics used were appropriate, especially in cases with carbapenemase-producing organisms. This could have influenced mortality, regardless of ADE decisions.
Author Response
- The study does not include illness severity scores at baseline (e.g. SOFA, APACHE II), which limits interpretation of mortality differences. Could higher mortality in the non-ADE group reflect more severe illness rather than the absence of ADE?
- This is something we frequently observe in daily practice. Unfortunately, we did not record those scores. In patients with more severe conditions, ADE might be deliberately avoided even when technically feasible, which could potentially strengthen the correlation between increased mortality rates and the absence of ADE. Following your suggestion, we have included this point within the text.
- In a patient with ESBL-producing E. coli bloodstream infection who was started on meropenem and remained hemodynamically unstable (e.g. on vasopressors) with hypoalbuminemia, clinicians might avoid switching to ertapenem despite in vitro susceptibility, due to concerns about suboptimal pharmacokinetics in critical illness. In such cases, was this considered a missed ADE?
- As we stated in the text, If ADE was theoretically feasible but not implemented, this is referred to as a missed ADE opportunity. In your scenario, we agree that it is hard to classify this as missed ADE. In such constrained situations, we did not classify it as a missed ADE. We, however, believe that a 7-day course of carbapenem therapy followed by a switch to TZP, particularly in the presence of clinical and laboratory improvement, should not be viewed with excessive concern. To support this, we’ve decided to extend the timeframe for ADE to seven days. We believe that within this extended period, a clinician would be able to identify a suitable moment for ADE, provided they are willing to do so.
- This is one of the important issues of the our study. Some colleagues believe that if there is ESBL positivity, then ADE is never We also emphasized this belief, whether it was right or not
- In ESBL-producing isolates susceptible to piperacillin-tazobactam, were cases that remained on carbapenem classified as missed ADE? Given MERINO data suggesting inferior outcomes with piperacillin-tazobactam compared to meropenem potentially due to OXA-1 co-production—was the lack of de-escalation in this study setting influenced by low OXA-1 prevalence?
- In ESBL-producing isolates susceptible to piperacillin-tazobactam (TZP), cases were considered missed ADEs if ADE was not applied despite being feasible. We are certainly aware about the intrinsic debate surrounding ADE in ESBL-positive cases. We believe that a 7-day course of carbapenem therapy followed by a switch to TZP, particularly in the presence of clinical and laboratory improvement, should not be viewed with excessive concern.
- In the MERINO study, the study drug was administered for the initial five days (at least), followed by discontinuation and a transition to oral treatment as an option. Therefore, MERINO study is not barrier for ADE practice. Our focus is on the outcomes observed within the first seven days. If the reviewer holds the perspective that ESBL positivity renders ADE impractical in all scenarios, we respect their position. However, we hope that the reviewer will recognize the importance of our study in fostering a discussion among readers about the implications of ESBL positivity and the absence of ADE in such cases. We were unable to analyze OXA-1 positivity, leaving us with insufficient evidence to draw conclusions about OXA-1 and the failure to switch to TZP.
- Thank you for your attention to those details, which we believe is one of the key aspects of our study. Following your suggestion, we have included MERINO study within the text.
- There is no discussion on whether the antibiotics used were appropriate, especially in cases with carbapenemase-producing organisms. This could have influenced mortality, regardless of ADE decisions.
- In Figure 2, we noted that 19 cases were excluded due to the use of inappropriate empirical antibiotics. To maintain the clarity of the study, these cases were omitted from the analysis.
- As we stated in the text, appropriate therapy was described as a treatment regimen incorporating at least one active drug, as outlined by EUCAST guidelines. For instance, the combination of meropenem with either amikacin or colistin was considered appropriate if the resistant bacteria displayed sensitivity to amikacin or colistin. Nevertheless, as we stated in the text, no ADE was used in bacteremic episodes caused by carbapenemase-positive isolates and multidrug-resistant Acinetobacter sp. (all of which were multidrug-resistant).
- There is no discussion on whether the antibiotics used were appropriate, especially in cases with carbapenemase-producing organisms. This could have influenced mortality, regardless of ADE decisions.
- In Figure 2, we noted that 19 cases were excluded due to the use of inappropriate empirical antibiotics. To maintain the clarity of the study, these cases were omitted from the analysis.
- As we stated in the text, appropriate therapy was described as a treatment regimen incorporating at least one active drug, as outlined by EUCAST guidelines. For instance, the combination of meropenem with either amikacin or colistin was considered appropriate if the resistant bacteria displayed sensitivity to amikacin or colistin. Nevertheless, as we stated in the text, no ADE was used in bacteremic episodes caused by carbapenemase-positive isolates and multidrug-resistant Acinetobacter sp. (all of which were multidrug-resistant).